# Generation of Orbital Angular Momentum Light by Patterning Azopolymer Thin Films

Temitope M. Olaleye, Maria Raposo * and Paulo A. Ribeiro

Laboratory of Instrumentation, Biomedical Engineering, and Radiation Physics (LIBPhys-UNL), Department of Physics, NOVA School of Science and Technology, Universidade NOVA de Lisboa, 2829-516 Caparica, Portugal; o.mary@campus.fct.unl.pt (T.M.O.); pfr@fct.unl.pt (P.A.R.)
* Correspondence: mfr@fct.unl.pt

**Abstract:** Orbital angular momentum (OAM) encoding is a promising technique to boost data transmission capacity in optical communications. Most recently, azobenzene films have gained attention as a versatile tool for creating and altering OAM-carrying beams. Unique features of azobenzene films make it possible to control molecular alignment through light-induced isomerization about the azo bond. This feature enables the fabrication of diffractive optical devices such as spiral phase plates and holograms by accurately imprinting a phase profile on the incident light. By forming azobenzene sheets into diffractive optical elements, such as spiral phase plates, one can selectively create OAM-carrying beams. Due to the helical wavefront and phase variation shown by these beams, multiple distinct channels can be encoded within a single optical beam. This can significantly increase the data transmission capacity of optical communication systems with this OAM multiplexing technique. Additionally, holographic optical components made from azobenzene films can be used to build and reconstruct intricate wavefronts. It is possible to create OAM-based holograms by imprinting holographic designs on azobenzene films, which makes it simpler to control and shape optical beams for specific communication requirements. In addition, azobenzene-based materials can then be suitable for integration into optical communication devices because of their reconfigurability, compactness, and infrastructure compatibility, which are the main future perspectives for achieving OAM-based technologies for the next generation, among other factors. In this paper, we see the possible use of azobenzene films in the generation and modification of OAM beams for optical communications through light-induced isomerization. In addition, the potential role of azobenzene films in the development of novel OAM-based devices that paves the way for the realization of high-capacity, OAM-enabled optical communication networks are discussed.

**Keywords:** orbital angular momentum; azobenzene; photoisomerization; optical communication





## 1. Introduction

In the modern digital landscape, the effective functioning of the internet and the operation of data centers heavily rely on the high-speed transmission of data across long distances through optical fiber networks. This is a result of the revolutionary advancement that the field of optical communication has undergone over the course of several decades. Photonics, encompassing a vast spectrum of applications, has played a pivotal role in not only transforming optical communication but also reshaping various research domains. The core concept of photonics revolves around the generation, manipulation, and detection of light beams in numerous ways [1]. These beams have been enabling innovations in optical communications, imaging, sensing, and beyond. As the demand for higher data transmission capacity keeps increasing exponentially and as the capacity crunch draws nearer [2], the quest to explore more innovative technologies and materials has become more important for researchers in this field.

One of the researched technologies is in Orbital Angular Momentum (OAM) based communication systems has remained in the limelight since its groundbreaking discovery

by Allen et al. in 1992 [3] following the initial theoretical research in 1991 [4]. Ongoing research has consistently demonstrated that harnessing the OAM of light presents a promising solution to significantly increasing the data transmission capacity of communication channels [5]. By exploiting unique phase profiles and orthogonal states inherent to OAM beams [6], these systems can not only enhance channel capacity but also effectively mitigate the channel capacity limitations of conventional systems reliant on the polarization of light [7,8]. Beams possess a spatial property characterized by a helical phase front. This has shown a lot of transformative capabilities in photonics, particularly because of its potential to enhance data transmission capacity by adding another degree of freedom for transmitting information in communication channels [9].

In addition to OAM systems, polarization-sensitive materials such as azobenzene [10] have emerged as a compelling candidate for research and investigation in the photonics field [11]. With the remarkable capacity of azobenzene for light-induced isomerization [12,13], the molecules of azobenzene films can dynamically change their orientation, resulting in the creation and modification of structured light such as OAM beams. Structured laser beams are laser beams that have been specifically manipulated to have predefined spatial intensity or phase distributions [14]. The photoisomerization capabilities of azobenzene, combined with the precision to imprint phase profiles onto incident light, enable azobenzene molecules to undergo reversible structural transformations as light passes through them [15]. With the use of optical devices such as spiral phase plates and holograms [16], azobenzene serves as a versatile tool, opening new methods for enhancing the fabrication of photonics devices [17–19], encoding information and information security [20] and transmitting data in optical communication systems.

This review article takes a more specific focus on the generation of OAM beams with the use of azopolymer thin films. Given a shortage of the existing literature in this sector at the time of writing, the use of azobenzene molecules for OAM synthesis represents a novel approach to the generation of structured beams such as OAM beams, improving our understanding of OAM and exploring its potential applications in photonics and optical communications.

*Objectives and Structure of the Paper*

In this review, we take a deep look into several useful characteristics of azobenzene materials and their utilization in the generation of OAM beams for various applications, with a particular focus on optical communications and photonic devices. This review starts with a description of the foundational principles of OAM, its significance in modern optical research, and the conventional techniques employed for its generation. Subsequently, we explore the unique characteristics of azobenzene materials, why they are suitable for generating structured light, and the process of producing azobenzene thin films. We then embark on a detailed journey through the innovative integration of azobenzene-based components to generate and modify OAM light beams. More importantly, we discuss recent research findings that shed light on the capabilities, challenges, and potential of this technology. Lastly, we conclude with a forward-looking perspective on the applications and future directions of azobenzene materials and the potential to revolutionize OAM-enabled optical communication networks.

## 2. Background, Generation, and Application of OAM Light

Electromagnetic waves are characterized by linear momentum and two distinct forms of angular momentum: spin angular momentum which is associated with circular polarization and the helicity of individual photons. It can take on values of $\pm 1$, corresponding to left- and right-handed circular polarization, and 0 for linear polarization.

On the other hand, OAM is associated with the spatial mode of a light beam with optical vortices [21,22], which are characterized by a twisted wavefront and a complex field amplitude [23]. The OAM of light is a new optical degree of freedom that arises from the spatial distribution of the wavefront and describes the rotation of the wave around

its propagation axis with a measure of intensity distribution and phase information [24]. OAM beams are characterized by a helical wavefront, which imparts a rotational motion to the beam around its propagation axis (Figure 1). Each OAM beam is quantized, meaning the beam carries a specific value of OAM denoted by the topological charge (TC) $\pm\ell$. The $\pm\ell$ represents the TC or the number of helical wavefronts the beam possesses in the clockwise (−) or anticlockwise (+) direction. Each photon in an OAM-carrying beam possesses an OAM of $\ell\hbar$, where $\hbar$ is the reduced Planck constant. The OAM value determines the number of helical wavefronts present in the beam. Unlike linear momentum or spin angular momentum, which are associated with polarization, OAM is a more intrinsic property of the optical field [25].

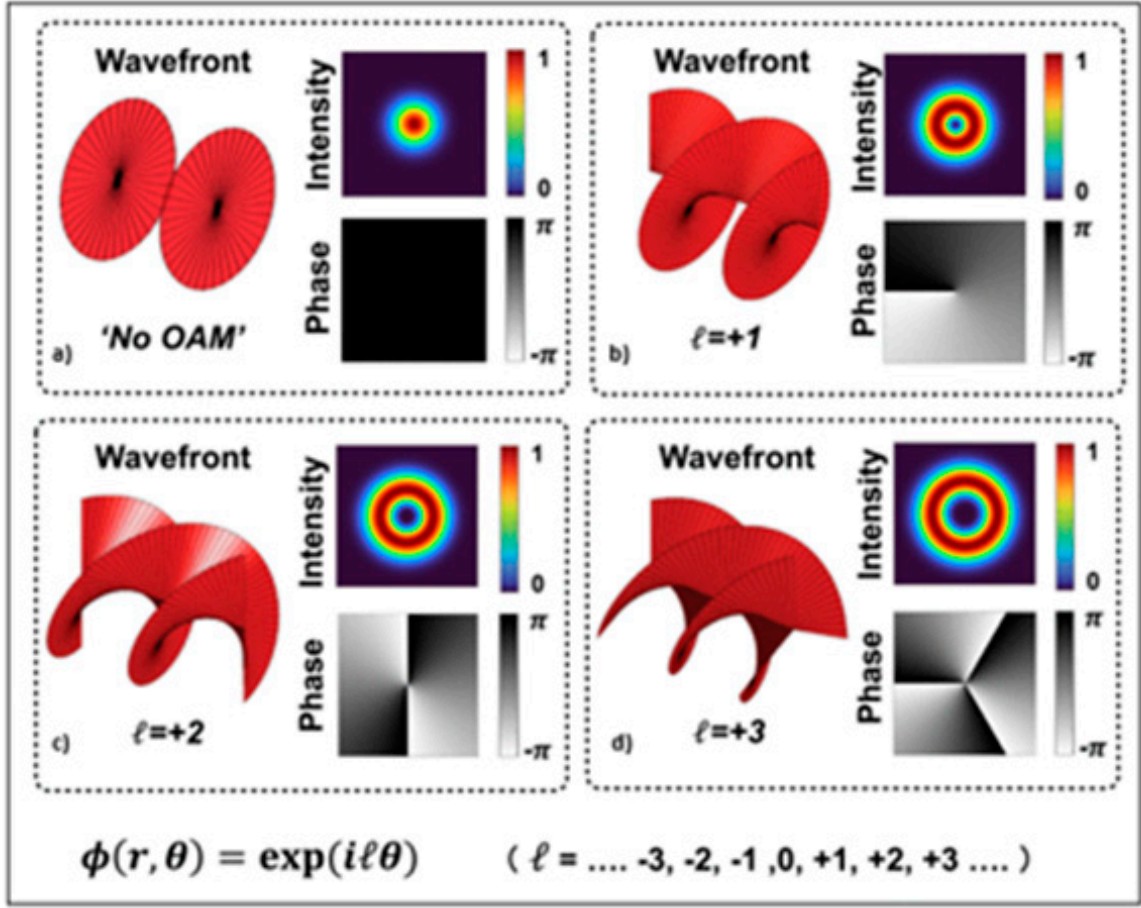

**Figure 1.** Wavefront, phase profile, and intensity profiles of OAM beams. (**a**) $\ell = 0$ represents a donut-shaped Gaussian beam with no twist/OAM in the wavefront; (**b**) $\ell = 1$, which defines one twist per wavelength; (**c**) $\ell = 2$, which defines two twists per wavelength; (**d**) $\ell = 3$, which defines three twists per wavelength. Reprinted from Ref. [26].

The study of OAM has primarily focused on Laguerre−Gaussian (LG) beams [27,28], which have well-defined values of OAM [29]. LG beams possess an azimuthal phase dependence of $\exp(i\ell\varphi)$, where $\ell$ also known as the TC is the beam's azimuthal mode number. It has a doughnut-shaped intensity profile determined by the beam's radial mode number ($p$). Owing to their helical wavefront, LG beams carry a quantized orbital angular momentum (OAM) of $\ell\hbar$ ($\hbar$ is the reduced Planck's constant) per photon, where the amount of OAM is dependent on $\ell$ [30]. These beams exhibit vortex-like structures (as seen in Figure 2). They have been extensively researched for their fundamental properties, methods of production, measurement, and applications in this reference [31].

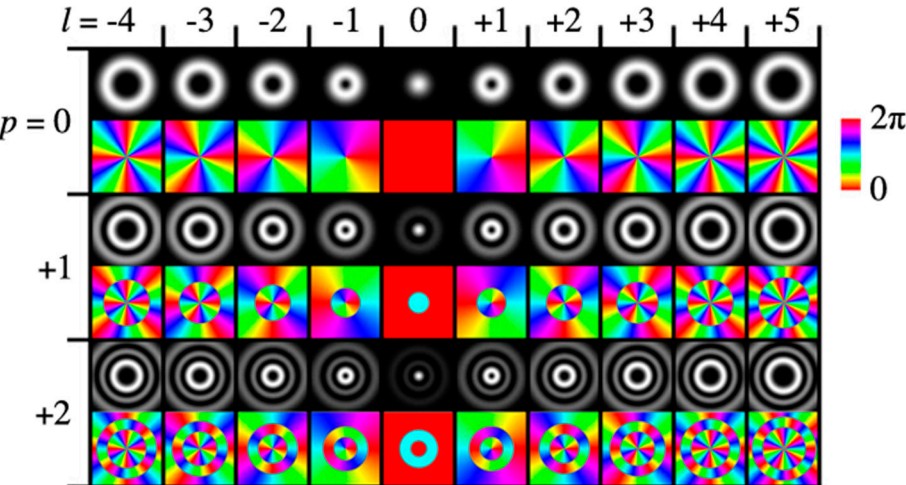

**Figure 2.** Intensity and phase profiles of LG modes. In the top row a combination of *p* = 0 and ℓ = 0 represents a Gaussian mode. The other modes represents LG. In each row, azimuthal mode number ℓ increases from left to right (−4 to +5), while the radial mode number *p* increases from top to bottom (0 to 2). Reprinted from [32].

OAM has been recognized as potentially useful for a vast and diverse range of applications, such as microparticle manipulation [33,34], trapped particle rotation [35], encoding of information [36], transfer of OAM to atoms [37], and some seen in Figure 3 [38]. Most especially, it has found utility in optical communications, where OAM multiplexing enables increased data transmission capacity [21].

Researchers have extensively delved into the generation and manipulation of OAM beams in free space using spatial-generating devices [39], such as cylindrical lenses [40,41], spiral phase plates [42,43], phase holograms [44,45], and spatial light modulators [46,47], and in optical fibers using fiber-generating [48,49] devices [31] which offer more advantages in terms of characteristics such as miniaturization, lower insertion loss, increased transmission distance, higher efficiency, and a reduction in external interference, which is lacking in the spatial generating method [50]. Converters such as fiber gratings [51], mode-selective couplers [52], photonic crystal fibers [53], and photonic lanterns [54] are attached to specially designed fibers to support OAM mode transmission. These methods have allowed the controlled production of light beams with specific OAM values. A detailed review of the background and overview of OAM beams, the fundamental concepts, various OAM generators, and the recent experimental and commercial applications of the OAM multiplexing technique in optical communications can be seen in the references [38,55]. Many of the above techniques require bulky configurations and multiple processing steps. To address this, researchers have proposed azobenzene films as a novel basis for OAM-based devices. Surface relief gratings (SRG) patterns on azobenzene films can be used to produce multiple OAM beams with varying topological charges and polarization states [56]. Using azobenzene also requires the use of the wavelengths or polarization of a writing light to control OAM beams. This will open new possibilities for fabricating compact and tunable OAM devices for diverse applications.

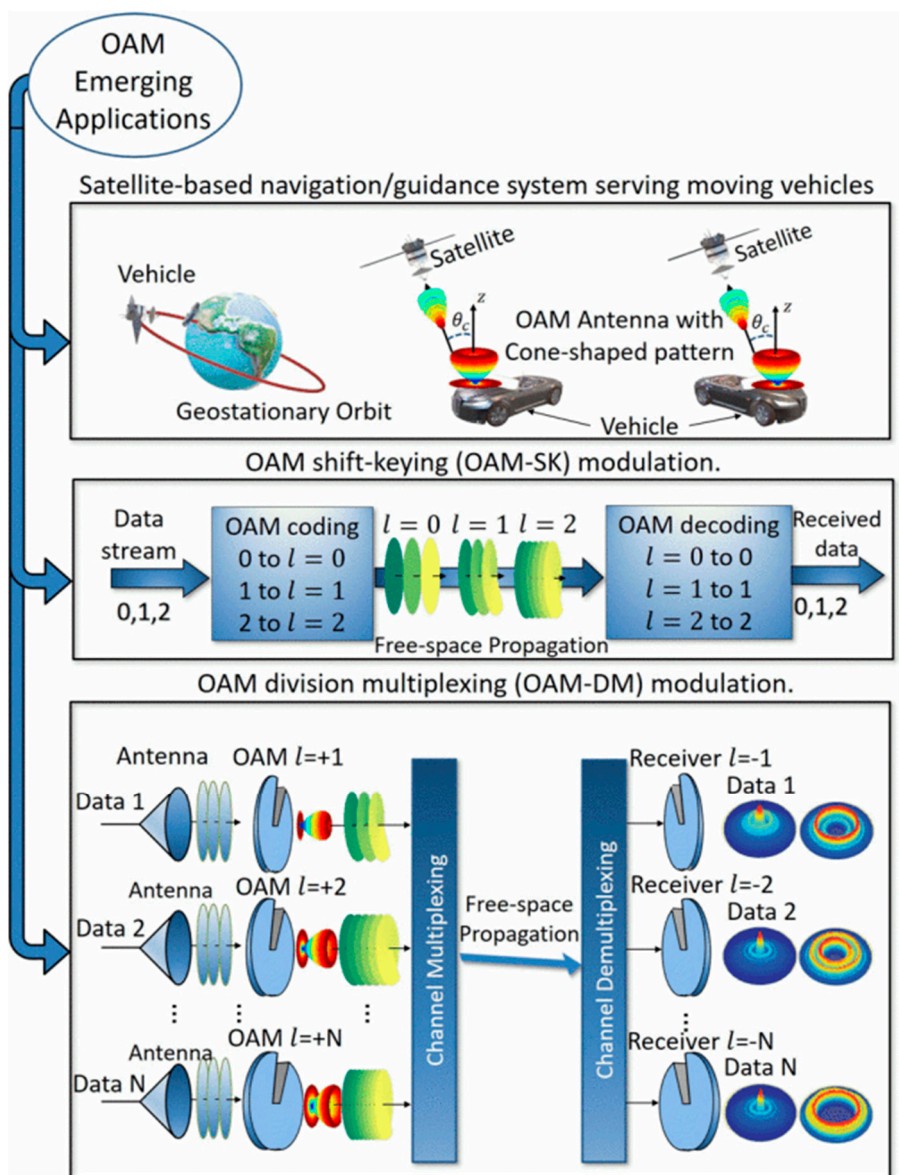

**Figure 3.** Emerging application of OAM beams. Reprinted from [38].

## 3. Azobenzene Materials

Azobenzene is an organic compound with the chemical formula $C_6H_5N=NC_6H_5$, and it consists of two benzene rings bound by a nitrogen−nitrogen (-N=N-) double bond, which is known as the azo group [57]. Azobenzene groups present two distinct structural forms: the *trans* and the *cis* form, as shown in Figure 4. These forms are distinguished in the spatial arrangement of their atoms; specifically, the orientation of the nitrogen−nitrogen (N=N) double bond rings on either side of the molecule are aligned in a straight line, which gives the molecule a longer and extended shape. However, in the *cis* form, the N=N double bond is bent, bringing the two benzene rings closer together and giving the molecule a more compact, bent structure.

A notable feature of azobenzene is its ability to undergo reversible isomerization, meaning it can switch between the *trans* and *cis* forms when exposed to light, particularly ultraviolet (UV) or visible light [58,59]. This is known as the photoisomerization process, during which the molecule can change between its *trans* and *cis* configurations upon exposure to specific light wavelengths [60]. While the *trans* form is thermodynamically preferred due to its stability, exposure to light or heat causes it to convert to the *cis* form, resulting in changes in its optical properties. The photochemical isomerization of azobenzene between

its *trans* and *cis* forms was first discovered in 1937 [61]. The photoisomerization process of the azobenzene group leads to the change of the spatial geometric arrangement, through the conversion of the isomer from *trans* to *cis* (*trans* → *cis*), induced by light absorption, or from *cis* to *trans* (*cis* → *trans*), induced by the action of light or heat. This process is associated with a n-π* transition of low absorption intensity in the visible region together with a higher intensity transition in the ultraviolet region.

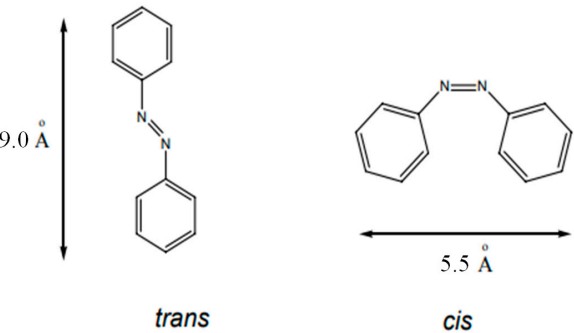

**Figure 4.** Schema of the *trans* and *cis* isomeric forms of azobenzene molecules or chemical groups.

In 1984, Todorov and his collaborators [62] described for the first time the formation of photoinduced birefringence by linearly polarized light in polyvinyl alcohol (PVA) mixed with the polar orange methyl chromophore. Results demonstrated that the photoinduced photoisomerization process gives rise to alignment of dipolar chromophores in the direction perpendicular to the polarization of the light electric field which consequently creates photoinduced birefringence in the medium. This process is a statistical approximation, since a chromophore preferentially absorbs light polarized along the axis of its dipole. The major axis of the molecule, with the probability of absorbing photons, is proportional to $cos^2\theta$, where $\theta$ is the angle between the direction of the electric field of light and the molecular dipole moment [63], as demonstrated in Figure 5a. Thus, the chromophores oriented in the direction of polarization of light absorb the light with a greater probability, unlike those which are not oriented perpendicularly and are not able to absorb this light and experience isomerization (see Figure 5b).

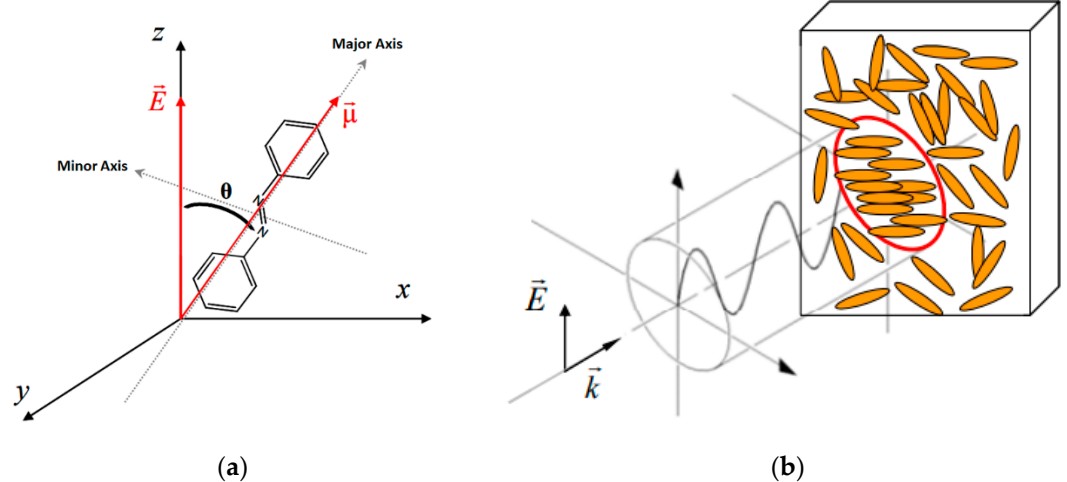

(**a**)　　　　　　　　　　　　　　　　　　　　　　　　(**b**)

**Figure 5.** (**a**) Schematic of the orientation of an azobenzene molecule relative to the electric field of light and its dipole moment $\vec{\mu}$; (**b**) schematization of the orientation of chromophores by the incidence of linearly polarized light: $\vec{E}$ represents the electric field vector, and $\vec{k}$ represents the wave vector. The region where the light falls tends to have chromophores oriented in the direction perpendicular to that of the light electric field.

If azobenzene is incorporated into a polymer chain, the photoisomerization reaction will occur in each azo group inserted in this chain [64]. This is a reversible reaction that does not involve the formation of secondary products [65], a so-called clean photoreaction. Photoisomerization was observed in solutions, in liquid crystals, in sol-gel systems, and in thin films of molecules with azobenzene groups or in mixtures of azobenzene with other molecules. Therefore, the azo group facilitates reversible photoisomerization, wherein the molecule can seamlessly transit between its *trans* and *cis* configurations upon exposure to specific light wavelengths. This inherent reversibility is fundamental for manipulating azobenzene materials in the context of structured light generation. The photoisomerization process in azobenzene involves the absorption of light energy, prompting the molecule to shift from its *trans* to *cis* configuration. This process exhibits high efficiency, with azobenzene displaying a notable quantum yield for photoisomerization. The *cis* configuration can then be converted back to the *trans* form, through either thermal or photochemical means, completing the reversible cycle. Azobenzene's capacity for light-driven control makes it an ideal candidate for crafting structured light.

In parallel to the photoinduced birefringence phenomenon observed in materials that contain azobenzenes, Natansohn and Rochon [66] in collaboration with Tripathy [67] found that when linearly polarized light is incident on the medium in the form of an interference pattern, not only does photoisomerization take place, but also changes in the medium volume are observed, which are translated by the formation of a relief grating. The inscription of the grids occurs by impinging two light beams in a given area on the surface of the film, so that an interference pattern is formed. Modulation amplitudes can be on the order of 100 to 1000 nm, and the grating period can be adjusted depending on the incident interference [68]. The creation of optical relief gratings involves the net transport of mass, a mobility that is only possible thanks to the *trans* → *cis* → *trans* photoisomerization processes of azobenzene chromophores. An image of a relief grid obtained by atomic force microscopy of self-assembled poly(carbonate chloride) dimethyldiallylammonium) (PDAC) with Congo red azopolymer (CR) is shown in Figure 6, adapted from an article by Tripathy [69]. Relief optical gratings have important applications in photonics, particularly in optical memories and holography [70].

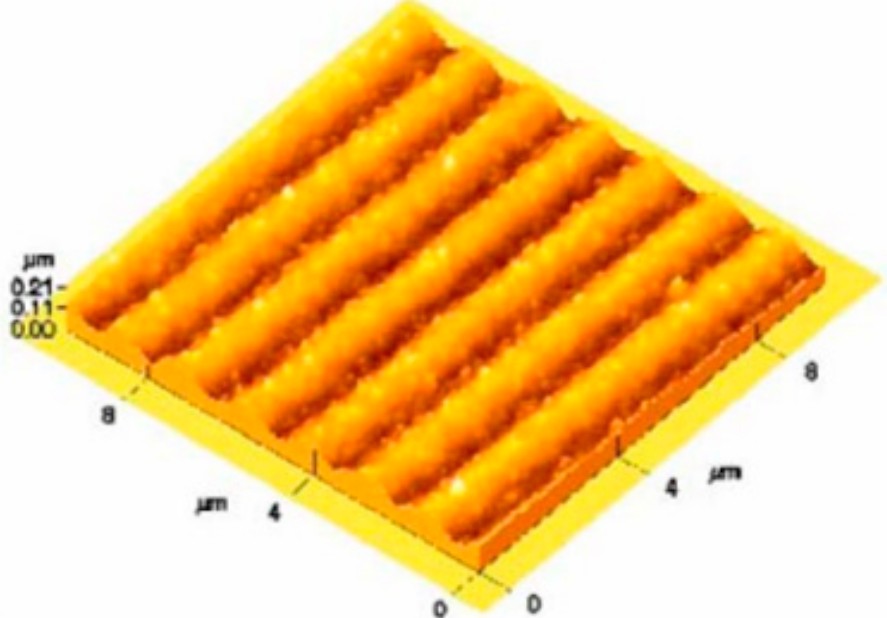

**Figure 6.** Atomic force microscopy (AFM) image of a relief grid of a PDAC/CR self-assembled film. Reprinted from Ref. [69].

Azobenzene-based molecules enable precise control in optics, photonics, and nanotechnology by creating light-responsive materials and switches. This property allows the controlled manipulation of molecular structure and properties of azobenzene molecules upon response to light exposure. They vary widely in profuse chemical structures and properties with a wide range of applications in the textile as dyes, chemicals, and materials. These derivatives are used to create functional materials and molecular switches that respond to light, enabling precise control over various systems and properties. This inherent light-induced structural change through either thermal or photochemical means is the fundamental for manipulating azobenzene materials in the context of structured light generation.

*Azobenzene Thin Films*

Due to the importance of optical properties that materials containing azobenzene can offer, it is essential to develop and optimize molecular structures of this type. The basic idea is to incorporate azobenzene into a host matrix to create a structure that maintains its photoisomerization capabilities. Several techniques such as casting, spin coating, the Langmuir−Blodgett (LB) technique, and the layer-by-layer (LbL) technique have been used to produce ultrathin films of azobenzene molecules. Here, one should highlight the LbL technique, which was developed in the 1990s by Decher et al. [71,72] based on adsorption at a solid−liquid interface. In this process, a monolayer of a positively charged polymer is initially obtained by dipping a hydrophilized substrate in a positively charged polyelectrolyte solution. Subsequently, the monolayer is washed to remove polyelectrolyte molecules that are not completely adsorbed. The solid support with a positively charged monolayer is immersed in a negatively charged polyelectrolyte solution to be adsorbed on a layer of negative polyelectrolyte. Following this step, a new wash is performed to remove unadsorbed molecules. This iterative procedure results in the formation of a bilayer composed of two oppositely charged polyelectrolytes. Repetition of this procedure leads to the gradual buildup of multilayers, ultimately resulting in the formation of a self-assembled film [73].

Compared to other known techniques, such as the Langmuir−Blodgett technique [68,74] or spin coating [75–77], the LbL technique has demonstrated to be an effective method for obtaining thin films. The advantage of this LbL technique compared to those mentioned lies in the fact that it is a simple, economical method compatible with large-scale production. It should be noted that the LbL technique also allows thickness control, which depends on fundamental factors that condition adsorption at a solid/liquid interface, such as ionic strength, concentration, pH, and temperature. The LbL technique can also be used on any type of substrate, regardless of its size or shape, and allows the use of water as a solvent, thus having potential to cause no harm to the environment. Initially, it was used only to produce and study oppositely charged polyelectrolyte structures, but quickly extended to functional molecules such as azopolymers [78]. Oliveira and collaborators carried out a review of the work already published on LbL azobenzene films and concluded that these azobenzenes are more difficult to photoisomerize [79]. However, in more recent studies, it was demonstrated with the use of higher temperatures, the orientation can be well achieved [80].

## 4. Azobenzene for OAM Generation and Manipulation

Azobenzene materials possess unique properties that render them ideal for creating and controlling structured light, particularly photons with OAM. In several fields such as optical communications, quantum computing, and nanophotonics, OAM manipulation is essential. The reversible photoisomerization of azobenzene aids the manipulation of molecular structure and facilitates the encoding of information within light's OAM. In addition, it allows quick modification of OAM states. This offers more flexibility in design options for photonic devices as well as increasing the response time for transmitting and processing high-speed information. The broad spectral range is another advantage of azobenzene isomerization, because it operates over a wide range of wavelengths such as in the UV,

visible, and near-infrared ranges [81]. This adaptability aids the manipulation of OAM and makes it suitable to implement with many optical systems. High-quality azopolymer surface patterns are easily controllable and can be imprinted, erased and reconfigured as needed. These patterns remain stable for several years under normal storage conditions. Azopolymers have significant potential for creating various photonic elements, including diffraction gratings [82], photonic crystals [83], nanostructured polarizers [84], plasmonic nanostructures [85,86], data storage units [87,88], and optical metasurfaces [89,90].

Scalability has been one of the major drawbacks of commercializing the use of OAM beams. However, by incorporating azobenzene into patterned structures, the creation of large-scale OAM modulation devices may be possible for high-speed OAM multiplexing in optical communication systems and OAM-based quantum information processing. Another advantage that is notable about azobenzene for OAM manipulation is the compatibility with existing technologies and materials commonly used in optics and photonics, and it can be incorporated into various host material structures, such as polymers [91] or liquid crystals [92,93], to form thin films or bulk materials. It also enables the creation of OAM-generating structures such as spatial light modulator (SLM) to modulate light's amplitude, phase, polarization, direction, and intensity [94,95]. To make an SLM with azobenzene polymers, a biphotonic holographic grating is used [96,97]. This grating is formed by the chromatic interference of light beams with different colors and polarizations [98], resulting in the creation of a physical grating structure/pattern with alternating regions of high and low intensity on the polymer surface. The photosensitive azobenzene polymer undergoes a change in its molecular structure in response to the light interference pattern where azobenzene molecules switch between different isomeric forms. This grating is then used as a spatial light modulator to diffract and control another light beam to change its phase and intensity [99]. The biphotonic holographic method in an azobenzene film and its reversible photoisomerization property is used for the storage of image/information [100]. In addition, liquid crystal systems that incorporates azobenzene enable the manipulation of refractive indices/birefringence and facilitate the development of optical elements that shape the OAM of light [101,102].

"Structured light" refers to light that can be controlled spatially in terms of its amplitude, phase, and polarization [103]. These parameters, together with the properties of an azobenzene film, must be manipulated to match certain criteria [14] for generation of structured light such as OAM beams:

i.   The azobenzene-enabled amplitude control plays a pivotal role in encoding information onto light by precisely modulating its intensity or brightness at different spatial points. This can be used to create patterns, enhance contrast, improve resolution and encode information. It is especially useful in microscopy and other optical applications. However, historically, lenses, prisms, apertures, and mirrors were the main static optical devices used in light manipulation, since accurate control over optical fields frequently required more complicated modifications [104]. One may accurately control the amplitude of a light beam as a first step toward better control, an idea that was crucial in the creation of holography. Amplitude masks were used in holography to simulate a "writing" laser beam which carries the information that is being encoded onto a holographic plate. Although clearly beneficial, this method is only able to use specified beam patterns [105].

ii.  Azobenzene materials also facilitate phase modulation by altering the timing or phases of different parts of a light wave. This process is integral to enabling beam shaping, producing structured wavefronts, and navigating light in desired directions in applications such as interference patterns, holography [106], and wavefront shaping within optical communication systems [107].

iii. Polarization control alters the orientation of the electric field vector of light. It is used in applications such as LCDs, 3D cinema for 3D effects, and optical communications for transmission of information. Azobenzene plays a valuable role in enabling the manipulation of light's polarization state for improving data-carrying capacity using

polarization-based multiplexing and demultiplexing techniques in optical communication systems.

The formation of structures in azopolymer thin films strongly depends on the polarization state of the illuminating laser beam [108], even though unpolarized light could also separate chiral molecules [109]. Hence, understanding the polarization of light is crucial for manipulating and utilizing light waves. Polarization describes the orientation of the oscillation of light waves, as they propagate in relation to the reference axis as seen in Figure 7a. In a process known as photoalignment [110], as polarized light oscillates, the molecules of azobenzene respond to the light exposure by aligning their orientation with it [1,111].

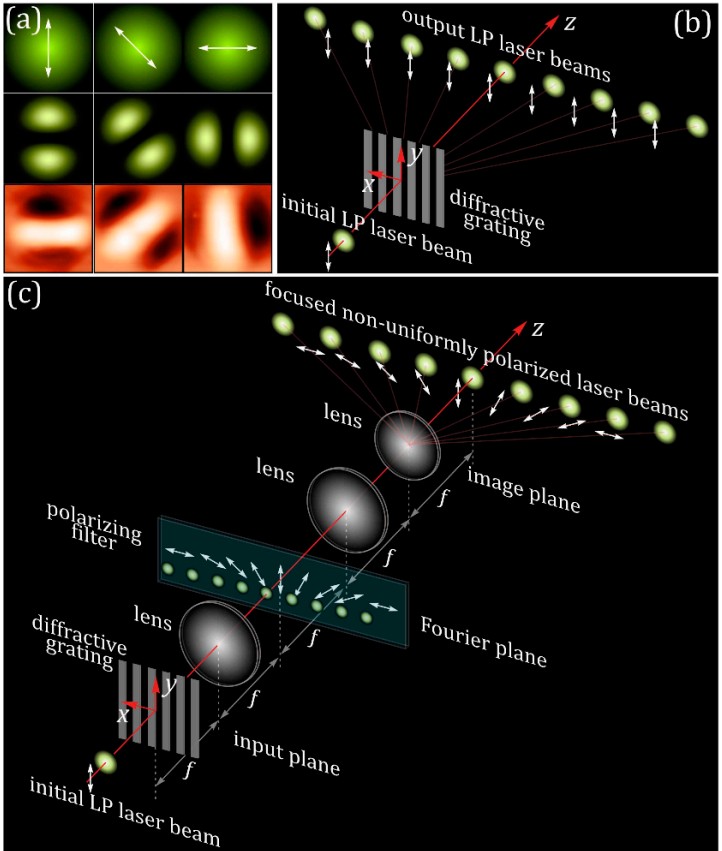

**Figure 7.** Implementation of polarization-sensitive patterning of azopolymer thin films. (**a**) Intensity distributions (top row) and the longitudinal components (middle row) of focused linearly polarized (LP) Gaussian laser beams with different polarization directions, as well as images of the microstructures formed in azopolymers thin films under the illumination of these beams (bottom row). (**b**) Splitting of a single LP Gaussian laser beam into a set of LP laser beams with a one-dimensional diffractive grating. (**c**) Principle of spatial polarization filtering and generation of a set of LP laser beams with different polarization directions using a 4-f optical system with a polarizing filter. Reprinted from [108].

To optimize the rate at which laser operations are performed, multiple laser beams with the same orientation are used. This is created by splitting a single laser beam using diffractive optical elements or metasurfaces as illustrated in Figure 7b. It is also possible to independently manipulate or adjust the polarization state of created light spots using a 4-f Fourier optical system, as shown in Figure 7c. With azopolymers, it is possible to control the profile and orientation of each of the formed structures, contrary to using interferometric lithography which does not allow this control [112,113]. Polarizing filters make it possible to encode information into incoming light and decode the information

using photosensitive materials such as azopolymers. Q-plates are useful for producing linearly polarized laser beams with a variety of polarization orientations, allowing for simultaneous, non-uniform laser patterning of thin azopolymer films [114].

### 4.1. Creation of Structured Beams by Spiral Mass Transport

To create structured beams, another important process to consider is spiral mass transfer [115,116]. During azobenzene photoisomerization, the transition between *cis* and *trans* configurations induces a controlled spiral or helical movement of molecules within the azopolymer film [117]. This movement is crucial for creating structured spiral-shaped patterns on the azopolymer thin film by polarization and controlled movement of the azo molecules [118]. This process or technique allows the modulation of both the amplitude and the phase of a light beam incident on the patterned thin film and induces a tailored OAM in the light [119,120]. A recent study [121] showed a novel technique for creating spiral structures on an azo-polymer film using circularly and linearly polarized beams in interference pattern processing. This research marked the first use of optical spiral radiation force in interference laser processing. Additionally, the study revealed the unexpected appearance of spiral relief patterns on a polymer film when exposed to focused LG beams with helical wavefronts and an optical vortex. These spiral patterns were found to be sensitive to both the vortex's TC and the wavefront's handedness, despite the unusual doughnut-shaped intensity profile of the LG beam (Figure 8). Further research is required to delve more into the method of spiral mass transfer.

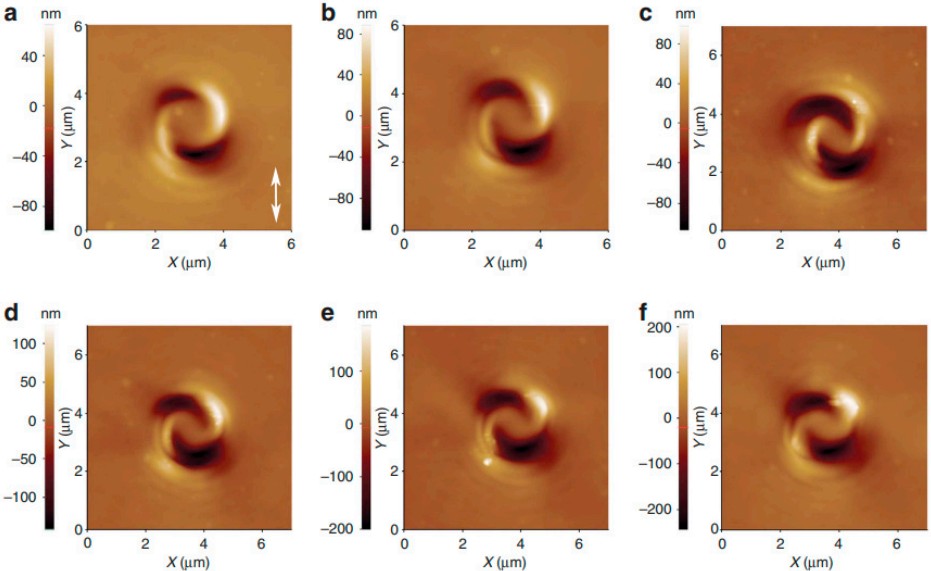

**Figure 8.** Spiral relief patterns obtained for different illumination doses from a Gaussian laser beam injected into a microscope. AFM images of the topographical structures were obtained with the varying illumination intensity and fixed time of exposure (and polarization direction) for topological charge $\ell$ = 10. The white arrow indicates the polarization direction. Different panels correspond to different values of the laser power injected in the microscope: (**a**) 15 mW; (**b**) 18 mW; (**c**) 21 mW; (**d**) 29 mW; (**e**) 41 mW; (**f**) 54 mW. Similar results were obtained for varying the time of exposure at a fixed intensity. Reprinted from [121].

To take advantage of the special qualities of azopolymers for optical and photonics applications, azobenzenes can be used as an active optical element [122] by directly altering their surface patterns in the processes of photoisomerization and mass transport; they can also act as templates or masks for the micro- or nanostructuring of other materials to produce a wide variety of photonic elements [123]. Both techniques make use of the special photoresponsive characteristics of azobenzene to fabricate SRGs [124] and produce structured light, such as OAM beams.

### 4.2. Azobenzene as an Optical Element for Generation of Structured Light

Azobenzene units undergo mass migration or transportation, as the N=N double bond switches between their two isomeric forms, *trans* and *cis*, in response to light patterns in the UV/visible wavelength range, resulting in surface relief patterns [125]. If periodic relief patterns are created on the material's surface, these can be exploited to control light. For example, to produce structured light, such as OAM beams, these relief patterns can be used to make diffractive optical elements (DOEs). Diffractive optical elements (DOEs) are specialized devices that are used to modify the amplitude or phase of light waves to produce specific patterns or images [126]. Spiral phase plates (SPPs) stand out among these DOEs, as they can produce optical vortices, which are light beams with helical wavefronts and a central phase singularity. These beams with OAM are generated by engineering the grating structure in such a way that it imparts a phase gradient across the diffracted orders, creating the characteristic helical phase profile associated with OAM beams. Surface gratings can be stacked using spacer layers, enabling the creation of three-dimensional chiral microstructures (Figure 9) [127] and diffractive azopolymer structures such as photonic crystals and optical nanomaterials [128].

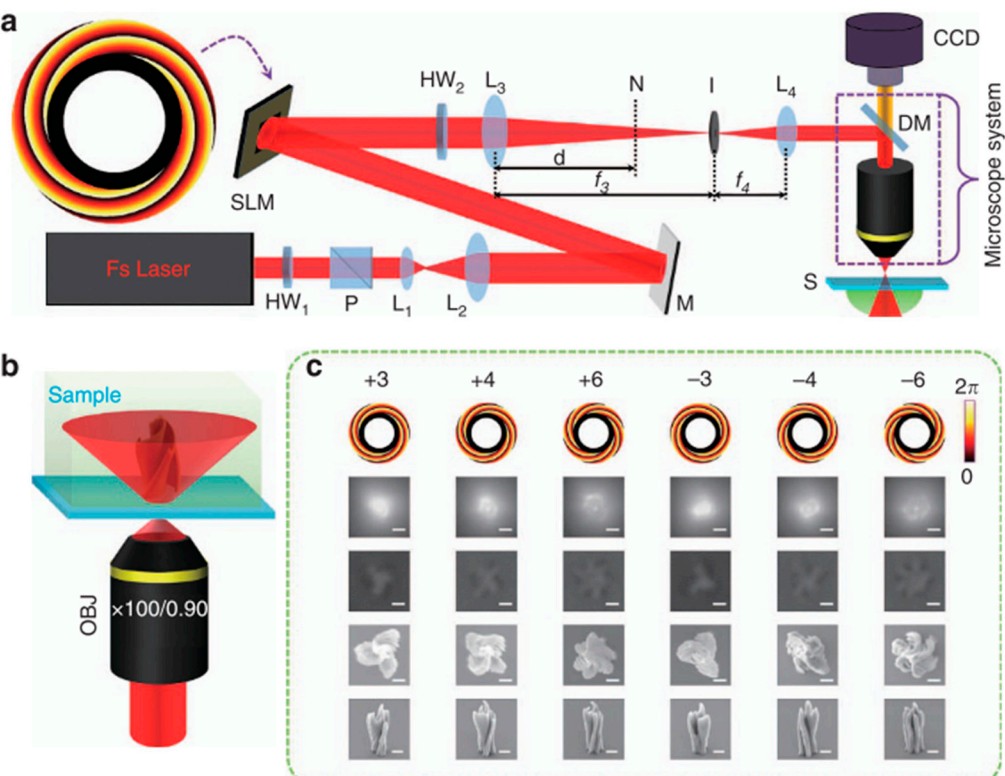

**Figure 9.** SLM-based experimental setup for generating 3D chiral microstructures in isotropic polymer by interfering beams of helical phase wavefronts and plane waves. In this experimental setup Fs Laser is a femtosecond laser, L1 and L2 are telescope lenses, HW1 and HW2 are half-wave plates, P is a polarizer, M is a mirror, SLM is a liquid-crystal spatial light modulator, I is an iris, L3 and L4 are lenses, DM is a dichroic mirror, S is the sample, and OBJ is a ×100 microscope objective. The square images below in the figure are SEM images of chiral microstructures achieved.Reprinted from [127].

Additionally, a diffraction grating that functions as a guided-mode resonant (GMR) filter with specific optical properties can be made by coating an azopolymer SRG with a layer of titanium dioxide [129]. Depending on the design and characteristics of the GMR structure, these filters interact with and modify the optical characteristics of the incident light by selectively transmitting or reflecting some wavelengths of light or its spectrum components, while suppressing others. By adjusting the filter's design parameters, such as the grating spacing, refractive indices, and layer thicknesses in response to various

wavelengths, structured light with specific spectral properties is produced [130,131], which can also potentially contribute to generating optical beams with OAM.

*4.3. Azobenzene as a Template/Mask for Generation of Structured Light*

Additionally, azobenzene molecules can serve as templates or masks for the micro- or nanostructuring of other materials to create OAM beams. In this method, an azo layer is applied to a substrate and exposed to light with a particular pattern or intensity distribution, such as an interference pattern or a Laguerre-Gauss beam. This way, the azobenzene layer functions as a template or mask for the underlying substrate. The beam triggers the photoisomerization process and mass transport in the azobenzene layer, causing a change in orientation and pattern/structure. The relief patterns made by the structured light are subsequently transferred from the azobenzene layer to the substrate. When irradiated with the proper incident light, this method enables the controlled micro- or nanostructuring of the substrate surface [132], which can then be used to produce OAM beams or other structured light patterns. This technique allows the design of unique surface patterns for desired optical effects [133].

Azobenzene can imprint OAM into a light propagating from a coherent laser source. A spatial light modulator (SLM) is used to design and create a phase pattern corresponding to the desired OAM mode in the laser beam. When the laser beam passes through the azobenzene film, the molecules of the azobenzene interacts with the laser light by changing their orientation, depending on the orientation and polarization of the light. The interferometric technique or the use of spiral phase plate detects and measures the presence and value of the OAM mode imprinted in the beam, after it interacts with thee azobenzene molecules [134]. This research describes how multi-spiral microstructures are generated by an interferometric approach in azopolymer thin films in the Mach−Zehnder interferometer. Spiral-shaped Intensity distributions/patterns are generated by the co-axial interference of OAM/optical vortex beams [135] with different topological charges (TC) $\ell$, and a gaussian beam with a spherical wavefront [136]. Due to the phase difference function of the two wavefronts, multi-spiral intensity patterns are generated. The fabrication method is simple to use, allowing for the profiles of the created microstructures to be changed without requiring changes to the optical configuration. The multi-spiral elements that are produced are compact and enable the creation of OVs at the microscale, with no restrictions on the quantity of spirals that can be formed (Figure 10a). A spiral-shaped intensity pattern is also formed using a spatial light modulator (Figure 10b) by interfering a focused optical vortex (OAM beam) [137], which is produced by a phase mask on the modulator, with a non-modulated portion of the original Gaussian beam.

The researchers further conducted another experiment using the Mach−Zehnder interferometer-based optical setup which consisted of a He-Ne laser source, beam splitters, lenses, diaphragms, and a video camera to generate OAM beams. The laser beam was focused on an azopolymer thin film which served as a template for the fabricated multi-spiral microstructures on a glass substrate. These structures were intended to modulate the incident laser beam by altering the phase and intensity of the incident beams, creating interference patterns. In the experiment, different multi-spiral structures with OVs of TCs +1, +2, and +3 were investigated. This resulted in an increase in the number of formed light spirals and reflected the value of the topological charge (TC) of the generated optical vortex (OV) beam. In the experimental setup in Figure 11a, it was noted that the thickness of the azopolymer thin film exceeded the heights of the microstructures, with the latter diminishing as TCs increased. No effect of the glass substrate on the properties of the patterned azopolymer films was seen; however, due to the reduction in the height of the formed patterns, the quality of the generated interference patterns reduced, as the number of formed spirals increased (Figure 11b). This is because the effectiveness of modulating the incident laser beam with microstructures relying on their height and the quality of interference patterns was directly impacted by the success of this modulation [138,139]. Hence, if the height of the microstructures is low, they may not adequately alter the phase

and intensity of the incident beam as intended. Consequently, the resulting interference patterns lacked high precision.

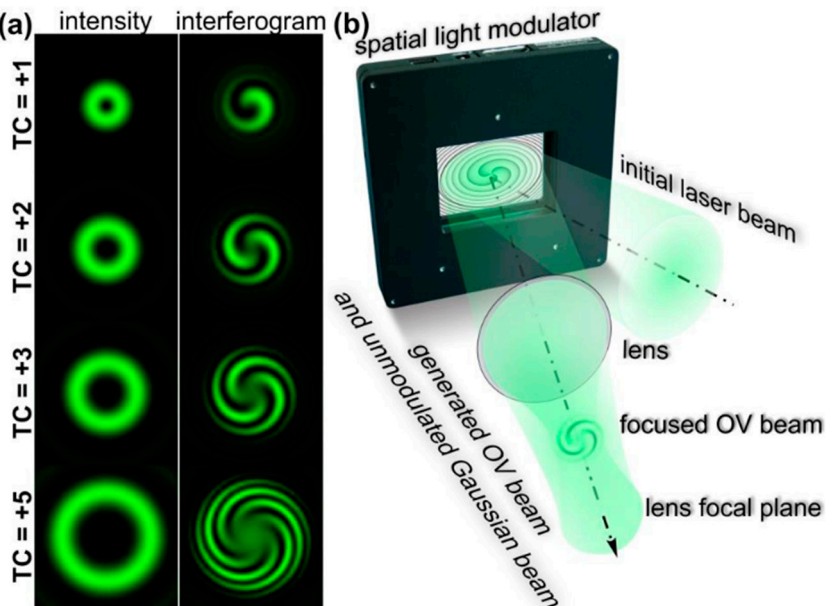

**Figure 10.** Generation of multi-spiral structures. (**a**) Spiral-shaped Intensity distribution/pattern generated by the interference of optical vortex beams with topological charges ($\ell = +1, +2, +3$, and $+5$) and a gaussian beam with a spherical wavefront; (**b**) spiral-shaped intensity pattern formed in a spatial light modulator by interfering a focused optical vortex (OAM beam), produced by a phase mask on the modulator with a non-modulated portion of a Gaussian beam. Reprinted from [134].

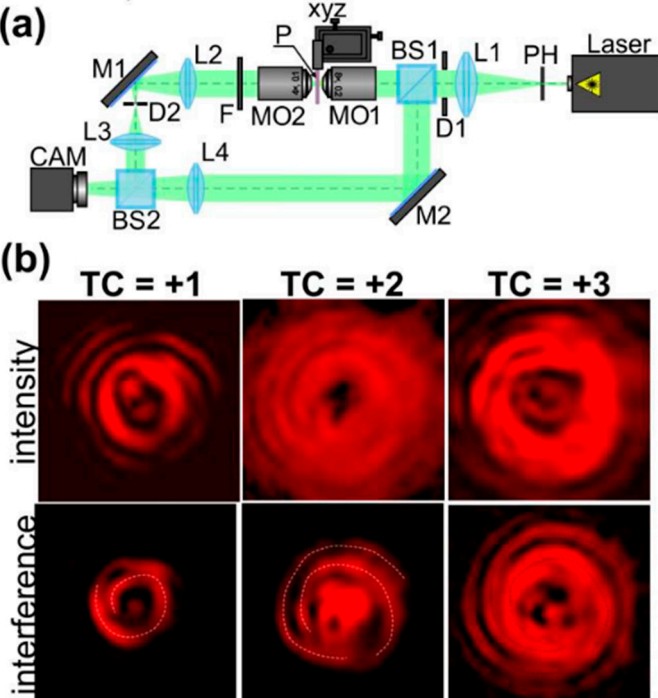

**Figure 11.** Production of OAM laser beams using multi-spiral microstructures created in azopolymer thin films using direct multi-spiral laser patterning. (**a**) The experimental setup to examine the light field created as a result of the diffraction of a linearly polarized Gaussian beam on the manufactured microstructures; (**b**) intensity distributions and interference fringes produced at a 30-micrometer separation from the azopolymer thin film surface. Reprinted from [134].

The result also demonstrated that the intensity pattern of the longitudinal component is strongly influenced by the state of polarization, and a more symmetrical distribution was detected with circular polarization [140].

## 5. Summary and Outlook

In this review, we have examined the properties of azobenzene-based materials for the creation and manipulation of OAM beams for optical communication applications. These materials have shown tremendous promise for developing the field of structured light, particularly in the production and manipulation of OAM and vortex beams, thanks to their distinctive photoresponsive characteristics. The review has touched on some of azobenzene's functions in optical transmission, with its effects on amplitude control, phase modulation, and polarization modification. Materials based on azobenzene have opened new possibilities such as the fabrication of a rewritable photonic crystal papers for patterned display applications [141], as well as for photonics optical communication, holography, and creation of cutting-edge optical components. Contrary to the current cumbersome setups of the other OAM generation or manipulation techniques, the compact nature of azobenzene-based materials makes them more attractive. It is possible to create novel data transmission, display, and imaging solutions by manipulating the phase, amplitude, and polarization states of light and encoding information into it. With further research and investigations, it is possible to see the integration of azobenzene materials into existing optical communication devices. Azobenzene materials play a pivotal role in these advancements, offering new opportunities for high-resolution imaging and data transfer.

Although azobenzene materials have many benefits, there are still problems such as scalability, environmental stability, and integration into practical devices for commercialization. OAM-enabled optical communication is very promising, and azobenzene-based materials will play a central role in enabling faster data transmission, higher-resolution displays, and improved optical systems. For future research, researchers are actively working on accurately tuning the parameters of the laser beam and exposure time during laser patterning and the possibilities of using phase masks of more complex diffractive optical elements for implementing the direct laser patterning of azo-polymer thin films [135].

In conclusion, azobenzene-based materials have become powerful and effective instruments for manipulating light and creating structured light such as OAM beams. Their potential in optical communication is evident, and ongoing research promises even more exciting developments. With their unique properties and versatility, they hold the key to unlocking new possibilities and reshaping the way we harness light for photonic applications and beyond.

**Author Contributions:** Conceptualization, M.R., T.M.O. and P.A.R.; writing—original draft preparation, T.M.O., M.R. and P.A.R.; writing—review and editing, P.A.R., M.R. and T.M.O.; supervision, P.A.R. and M.R.; funding acquisition, P.A.R. and M.R. All authors have read and agreed to the published version of the manuscript.

**Funding:** This research was funded by the Portuguese National Funding Agency (FCT-MCTES), UIDB/04559/2020 (LIBPhys), and UIDP/04559/2020 (LIBPhys).

**Conflicts of Interest:** The authors declare no conflict of interest.

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
