# Peer review of "Generation of Orbital Angular Momentum Light by Patterning Azopolymer Thin Films"

_photonics, doi:10.3390/photonics10121319_

Round 1
Reviewer 1 Report
Comments and Suggestions for Authors
Dear Authors!
All my comments to the paper are in the attached file.

Dear Authors!
All my comments to the paper are in the attached file.
Author Response
Response to the Comments
We thank the reviewers for the comments and suggestions that we think are properly addressed in the present form of the manuscript. We believe that corrections made fully accomplished reviewers comments and suggestions and improved its overall quality has been achieved.
Reviewer #1
Dear Authors! The paper is interesting and actual, I think that it will be read by scientists working in corresponding fields. At the same time, some errors are present in the text, and I propose to make additional editing with the aim to improve the paper. Particularly, I would like to focus the attention at the following:
Point 1. Line 250: “…incorporate azobenzenes into a system to create a system …” Too many “systems” in one sentence. I propose to make the expression more accurate.
Answer: The authors appreciate your feedback. The suggested modification has been addressed and the sentence revised to enhance clarity and avoid repetition.
Point 2. Lines 258 and 259: “ … now of opposite charge negative charge …” I do not understand this phrase, and think that other readers will also have some difficulty in understanding.
Answer: The authors acknowledge the confusion in Lines 258 and 259. The phrase has been revised for better clarity to ensure readers have a clear understanding of the intended meaning.
Point 3. Line 307: “…interference of two light beams with different colors and polarizations …” Is the interference possible of two light beams with different colors (i.e., with different wave lengths)? I do not know and ask the Authors to find some paper with example of such interference.
Answer: Thank you for pointing this out. Interference between light waves of different colors is indeed possible. This phenomenon is known as "chromatic interference". This usually arises from dispersion, differential refraction, or color fringing and leads to the creation of a physical grating structure/pattern with alternating regions of high and low intensity. The pattern formed is known as biphotonic holographic grating. When placed on the surface of a photosensitive material like azobenzene. The photosensitive azobenzene polymer undergoes a change in its molecular structure in response to the light interference pattern where the azobenzene molecules switch between different isomeric forms. This grating can be used as a spatial light modulator to control the phase and intensity of another incident light. In the experiment referenced in the manuscript, the interference of two light waves was done; a Gaussian laser beam with a spherical wavefront and an OV beam with a non-zero TC to generate multi-spiral light patterns for the purpose of manipulating the phase difference and amplitudes of the resulting pattern and generate a structured beam.
Point 4. Lines 316 and 317: “ … for generation of structured light like OAM …” The sentence can be re-written as “ … for generation of structured light like orbital angular momentum …” I am not sure that light can be like momentum and propose to change the expression.
Answer: The authors thank you for your suggestions on rephrasing the sentence. The phrase has been revised to “…for generation of structured light like OAM beams” to enhance clarity. This adjustment aims to ensure that the focus is on OAM beams rather than just OAM.
- Lines 344 – 350. Such description of polarization is appropriate in a tutorial on optics but can be omitted in a scientific paper.
Answer: The authors appreciate your input, and we agree that the lengthy explanation of polarization included in the manuscript could be unnecessary for a scientific work. We have updated the manuscript with the assumption that the readers have a basic comprehension of the concept.
This list can be extended, but I want to stop here and suggest to the authors to read the article again and improve the text.
Answer: We deeply appreciate your insightful remarks and valuable feedback. We have read the manuscript again to make sure the updated version is more concise and understandable for the readers. Thank you.
Reviewer 2 Report
Comments and Suggestions for Authors
Olaleye et al present a review about the generation of OAM beams with the use of Azopolymer thin films. This article is generally organized, however, there are several issues that need to be clarified before I can recommend publication of this manuscript.
1. The order of the caption in the figures is incorrect, line 214 Figure 2. Both the captions and manuscript should be carefully checked to avoid unnecessary errors.
2. The resolution of the images should be improved, such as Figures 1, 4, and 2 (line 214).
3. In Figures 4 and 6, it is not appropriate to include only one image from the literature. More relevant data from the literature should be presented.
4. The schematic diagram and mechanism of the OMA device should be specifically explained in a figure.
5. In Figure 3, the author should provide the changes in molecular steric configuration before and after cis-trans isomerization of azobenzene mesogen.
6. Azobenzene patterned display is one of the important applications. The following literature should be cited: ACS Appl. Mater. Interfaces 2021, 13, 12383-12392; Angew. Chem., Int. Ed. 2020, 59, 4035-4042.
Comments on the Quality of English LanguageNone
Author Response
Response to the Comments
We thank the reviewers for the comments and suggestions that we think are properly addressed in the present form of the manuscript. We believe that corrections made fully accomplished reviewers comments and suggestions and improved its overall quality has been achieved.
Reviewer #2
Olaleye et al present a review about the generation of OAM beams with the use of Azopolymer thin films. This article is generally organized, however, there are several issues that need to be clarified before I can recommend the publication of this manuscript.
Point 1. The order of the caption in the figures is incorrect, line 214 Figure 2. Both the captions and manuscript should be carefully checked to avoid unnecessary errors.
Answer: We appreciate your careful review and thank you for bringing our attention to the order of the captions in the figures. In the revised manuscript, we have ensured that the captions are correctly ordered.
Point 2. The resolution of the images should be improved, such as Figures 1, 4, and 2 (line 214).
Answer: Thank you for your suggestion regarding the resolution of the images. We have enhanced the resolution of these images to provide clearer visual representations for the readers in the revised manuscript.
Point 3. In Figures 4 and 6, it is not appropriate to include only one image from the literature. More relevant data from the literature should be presented.
Answer: We appreciate your feedback concerning the figures. We understand the importance of providing more relevant data from the literature. In the revised manuscript, we have included additional data and images from the literature to improve the comprehensiveness of our review.
Point 4. The schematic diagram and mechanism of the OAM device should be specifically explained in a figure.
Answer: Thank you for your suggestion. We would like to highlight that Figures 9, 10, and 11 in the manuscript have been meticulously designed by the referenced authors to illustrate the mechanisms of the OAM-generating device. These figures provide a detailed process involved in OAM generation. Hence, we did not want to add an additional schematic which might lead to redundancy and potential confusion for the readers. However, we are open to further refining and enhancing the existing figures to improve the clarity of the current figures or any additional information you believe would enhance the reader's understanding.
Once again, we appreciate your thorough review and remain committed to ensuring the clarity and coherence of our manuscript.
Point 5. In Figure 3, the author should provide the changes in molecular steric configuration before and after cis-trans isomerization of azobenzene mesogen.
Answer: Thank you for pointing out this omission. In the figure3, now 4, one intends to show only the azobenzene group in cis and trans forms and their respective sizes. As different azobenzene molecules can be synthetized and as the azobenzene systems are not necessarily liquid crystals, so mesophase may not be form. This was the reason because the changes in molecular steric configuration before and after cis-trans isomerization of azobenzene mesogen are not presented.
Point 6. Azobenzene patterned display is one of the important applications. The following literature should be cited: ACS Appl. Mater. Interfaces 2021, 13, 12383-12392; Angew. Chem., Int. Ed. 2020, 59, 4035-4042.
Answer: The authors thank you very much for the suggested literature reference. We have incorporated the citation in section 5 of the manuscript to enhance the discussion on the applications of azobenzene.
Reviewer 3 Report
Comments and Suggestions for Authors
This manuscript provides a comprehensive and informative review of the properties and applications of azobenzene-based materials for the generation and manipulation of structured light, particularly orbital angular momentum (OAM) beams. The authors effectively highlight the unique characteristics of azobenzene materials that make them promising tools for optical communication and photonic applications. The review is well-structured, offering clear explanations of concepts and a logical flow of information. While discussing the strengths of azobenzene materials, the manuscript also acknowledges challenges, such as scalability and environmental stability. Moreover, the authors hint at ongoing research, hinting at a promising future for this field. The manuscript can be recommended for publication in Photonics. There are still some comments for the authors to consider for improvement.
1. Please check the section titles. In Section 1 (Introduction), there is not a subsection “1.1”. There are two “Section 2”: “2. Background, Generation and Application of OAM light” and “2. Azobenzene materials”. For the unnumbered sections “Azobenzene as Optical Element for Generation of Structured Light” and “Azobenzene as Template/Mask for Generation of Structured Light”. Could the authors consider to mark those as subsections?
2. In the text, the abbreviation “orbital angular momentum (OAM)” is repetitively defined three times in Line 50-62. Additionally, please use “OAM” after the first time of abbreviation definition.
3. The authors mention the Laguerre-Gauss (LG) beams. As a comprehensive review, I suggest the authors to introduce more about LG beams and their specific characteristics to help readers who may not be quite familiar with that.
4. Could the authors provide some more detailed explanations of spiral mass transfer to help readers understand its significance for the generation of OAM beams more effectively?
5. The discussion about the reduction in the quality of generated interference patterns as the number of formed spirals increases is interesting. However, it would be beneficial to explain why this happens in more detail.
6. Please the authors carefully check the manuscript again to avoid some typos and small errors, e.g., “such as OAM beams,” in Line 403. The comma should be a full stop.
Comments on the Quality of English LanguageMinor editing of English language required
Author Response
Response to the Comments
We thank the reviewers for the comments and suggestions that we think are properly addressed in the present form of the manuscript. We believe that corrections made fully accomplished reviewers comments and suggestions and improved its overall quality has been achieved.
Reviewer #3
This manuscript provides a comprehensive and informative review of the properties and applications of azobenzene-based materials for the generation and manipulation of structured light, particularly orbital angular momentum (OAM) beams. The authors effectively highlight the unique characteristics of azobenzene materials that make them promising tools for optical communication and photonic applications. The review is well-structured, offering clear explanations of concepts and a logical flow of information. While discussing the strengths of azobenzene materials, the manuscript also acknowledges challenges, such as scalability and environmental stability. Moreover, the authors hint at ongoing research, hinting at a promising future for this field. The manuscript can be recommended for publication in Photonics. There are still some comments for the authors to consider for improvement.
Point 1: Please check the section titles. In Section 1 (Introduction), there is not a subsection “1.1”. There are two “Section 2”: “2. Background, Generation and Application of OAM light” and “2. Azobenzene materials”. For the unnumbered sections “Azobenzene as Optical Element for Generation of Structured Light” and “Azobenzene as Template/Mask for Generation of Structured Light”. Could the authors consider to mark those as subsections?
Answer: The authors thank you for your awareness of these numbering mistakes and have done them properly.
Point 2. In the text, the abbreviation “orbital angular momentum (OAM)” is repetitively defined three times in Line 50-62. Additionally, please use “OAM” after the first time of the abbreviation definition.
Answer: Thank you, this is true, and it has been adjusted accordingly throughout the manuscript.
Point 3. The authors mention the Laguerre-Gauss (LG) beams. As a comprehensive review, I suggest the authors to introduce more about LG beams and their specific characteristics to help readers who may not be quite familiar with that.
Answer: Thank you for this suggestion. We mentioned the LG beams briefly and added some references that discussed them in detail. But as you suggested for readers who may not be familiar with the LG beams, we have added more content about this in the manuscript.
Point 4. Could the authors provide some more detailed explanations of spiral mass transfer to help readers understand its significance for the generation of OAM beams more effectively?
Answer: More explanation has been added on however, further research is required to delve more into the method of spiral mass transfer.
Point 5. The discussion about the reduction in the quality of generated interference patterns as the number of formed spirals increases is interesting. However, it would be beneficial to explain why this happens in more detail.
Answer: Yes, indeed it is. This is because the effectiveness of modulating the incident laser beam with microstructures relies on their height and the quality of interference patterns is directly impacted by the success of this modulation. Hence, if the height of the microstructures is low, they may not adequately alter the phase and intensity of the incident beam as intended. Consequently, the resulting interference patterns will lack high precision. More details have been added to the manuscript.
Point 6. Please the authors carefully check the manuscript again to avoid some typos and small errors, e.g., “such as OAM beams,” in Line 403. The comma should be a full stop.
Answer: The authors thank you for noticing this typo error. The manuscript was read again, and all typo errors have been corrected.